# Beta-Amyloid Peptide in Tears: An Early Diagnostic Marker of Alzheimer’s Disease Correlated with Choroidal Thickness

**DOI:** 10.3390/ijms24032590

**Published:** 2023-01-30

**Authors:** Magda Gharbiya, Giacomo Visioli, Alessandro Trebbastoni, Giuseppe Maria Albanese, Mayra Colardo, Fabrizia D’Antonio, Marco Segatto, Alessandro Lambiase

**Affiliations:** 1Department of Sense Organs, Sapienza University of Rome, 155, Viale del Policlinico, 00161 Rome, Italy; 2Department of Human Neurosciences, Sapienza University of Rome, 00185 Rome, Italy; 3Department of Biosciences and Territory, University of Molise, 86100 Campobasso, Italy

**Keywords:** Alzheimer’s disease, biomarkers, tears, choroidal thickness, beta-amyloid, APP, p-tau

## Abstract

We aimed to evaluate the diagnostic role of Alzheimer’s disease (AD) biomarkers in tears as well as their association with retinal and choroidal microstructures. In a cross-sectional study, 35 subjects (age 71.7 ± 6.9 years) were included: 11 with prodromal AD (MCI), 10 with mild-to-moderate AD, and 14 healthy controls. The diagnosis of AD and MCI was confirmed according to a complete neuropsychological evaluation and PET or MRI imaging. After tear sample collection, β-amyloid peptide Aβ1-42 concentration was analyzed using ELISA, whereas C-terminal fragments of the amyloid precursor protein (APP-CTF) and phosphorylated tau (p-tau) were assessed by Western blot. Retinal layers and choroidal thickness (CT) were acquired by spectral-domain optical coherence tomography (SD-OCT). Aβ1-42 levels in tears were able to detect both MCI and AD patients with a specificity of 93% and a sensitivity of 81% (AUC = 0.91). Tear levels of Aβ1-42 were lower, both in the MCI (*p* < 0.01) and in the AD group (*p* < 0.001) when compared to healthy controls. Further, Aβ1-42 was correlated with psychometric scores (*p* < 0.001) and CT (*p* < 0.01). CT was thinner in the affected patients (*p* = 0.035). No differences were observed for APP-CTF and p-tau relative abundance in tears. Testing Aβ1-42 levels in tears seems to be a minimally invasive, cost-saving method for early detection and diagnosis of AD.

## 1. Introduction

The identification of new, less invasive, and cost-saving methods for the early detection of Alzheimer’s disease (AD) currently represents one of the main challenges in research. At present, the most well-established biomarkers for AD include the evaluation of beta-amyloid peptides (Aβ1-42), total tau (t-tau), and phosphorylated-tau (p-tau) in the cerebrospinal fluid (CSF) as well as the identification of amyloid deposition in the PET scans [1]. However, these methods are expensive, invasive, and cannot be easily performed in developing countries. In recent years, there were increasing efforts to find ocular biomarkers in AD based on the evidence of the early onset of visual symptoms in AD, and on the easy accessibility of ocular structures both in the sampling of biological material and for the direct visualization of the neuronal and vascular structures [2]. Previous studies identified several retinal and vascular abnormalities including choroidal or retinal nerve fiber layer thinning, and amyloid precursor protein (APP) deposits in the retina [3,4,5]. However, to date, no definitive evidence is available [6]. More recently, the detection of biomarkers for AD in tears has been investigated by a few authors [7]. At present, only one study investigated the diagnostic role of classical AD biomarkers in the tears of patients with dementia, however, with inconclusive results [8]. The purpose of this study is to assess the diagnostic role of classical AD biomarkers in the tears of a cohort of AD patients with an age- and sex-matched control group comparison. In parallel, we aimed at evaluating the correlation between AD biomarkers in tears with retinal and choroidal microstructures.

## 2. Results

According to the inclusion and exclusion criteria, 66.1% of the subjects assessed for eligibility were enrolled: 25 patients with MCI due to AD or mild-to-moderate AD, and 16 healthy controls. After enrollment, 4 affected patients and 2 healthy controls were further excluded for insufficient tear sample collection. Finally, 11 patients with MCI, 10 patients with mild-to-moderate AD, and 14 healthy patients were included in this study. A detailed flowchart of patients’ enrollment and further exclusions is reported in Figure 1. 

Overall, as shown in Table 1, no substantial differences were found between all the affected patients and controls when comparing sex (female 57.1%), age (71.7 ± 6.9, range 57–85), and systemic diseases. Conversely, as expected, MMSE was significantly lower and ADAS-Cog considerably higher in the affected patients.

The same characteristics were analyzed in the three groups of study subjects (Table 2).

Ophthalmological data were tested for the differences between affected patients and healthy subjects and are reported in Table 3.

As shown, in both groups, patients were mostly phakic and had a compatible refractive error (SEQ). Concerning SD-OCT findings, the analysis of pRNFL as well as central foveal thickness, NFL and GCP-IPL showed high compatibility between affected patients and controls. However, when comparing choroidal thickness (CT), affected patients showed a significant thinning compared to the control group (*p* = 0.035, diff. 39.1, CI 3.0 to 75.3). In Table 4, the same characteristics were analyzed within the three groups showing low compatibility of CT between AD and control group (*p* = 0.069, contrast −49.4, CI −101.9 to 3.2) after one-way ANOVA (F(2,32) = 2.78, *p* = 0.077).

### 2.1. Tear Sample Analysis of Alzheimer’s Biomarkers

Aβ1-42, APP-CTF, and p-tau were detected in the tear samples of all included subjects. Results from the tear sample evaluation are reported in Table 5.

Concerning the evaluation of Aβ1-42 by ELISA, a Kruskal–Wallis rank test (χ2(2) = 17.908, *p* < 0.001) highlighted a substantial difference among the three groups. Specifically, Dunn’s pairwise comparison revealed that Aβ1-42 concentrations were significantly lower in MCI (z = 2.79, *p* = 0.008) and AD (z = 4.07, *p* < 0.001) groups when compared to the control group, while no differences were found between the MCI and AD groups (z = 1.28, *p* = 0.298). In Figure 2, strip plots with Aβ1-42 values and distribution among groups are shown.

With regard to the p-tau relative abundance as evaluated by Western blot, after the Kruskal–Wallis rank test (χ2(2) = 2.712, *p* = 0.257), no differences among the three groups (AD, MCI, and control group) were highlighted. In Figure 3, strip plots with p-tau values and distribution among groups are shown.

Regarding the estimation of APP-CTF fragments, Western blot evaluation revealed the presence of a predominant immunoreactive band, with a molecular weight of approximately 25 kDa. Notably, the detection of 25 kDa CTF fragments was already observed in CSF from AD patients [9]. In this context, we performed a densitometric analysis of the 25 kDa CTF and we found a significant difference between groups as determined by one-way ANOVA (F(2,32) = 7.95, *p* = 0.002). A Tukey post hoc test highlighted that APP-CTF was lower in MCI patients when compared to AD patients. However, no significant differences were observed when comparing MCI patients and AD patients to the control group. In Figure 4, strip plots with APP-CTF relative abundance values and distribution among groups are shown.

Representative Western blot of p-tau and APP-CTF are shown in Figure 5a,b, respectively.

### 2.2. Neurological and Ophthalmological Correlations

When testing Spearman’s rank correlations, Aβ1-42 was found to be correlated with the affected group (rho = −0.69, *p* < 0.001), MMSE (rho = 0.60, *p* < 0.001), ADAS-Cog (rho = −0.63, *p* < 0.001), and CT (rho = 0.49, *p* = 0.003). A post hoc analysis (Figure 6) to investigate the relationship between Aβ1-42 and CT was performed using a non-parametric kernel regression (R2 = 0.16) with an estimated effect of 0.56 (CI 0.21 to 0.99, *p* = 0.015).

Conversely, Pearson’s correlations were tested between APP-CTF and all the neurological and ophthalmological variables, and no significant correlation was found. Similarly, Spearman’s rank correlations were tested for p-tau with no significant results. After a logistic regression analysis (cons. −0.47, CI −15.36 to 14.42, R2 = 0.55), corrected by age, sex, and all the tested AD biomarkers as well as CT, lower levels of Aβ1-42 were retained as the only predictor for MCI and AD (coeff. −0.05, CI −0.10 to −0.02).

A receiver operating characteristic (ROC) curve (Figure 7) was run to test the diagnostic ability of Aβ1-42 in the detection of AD (AUC 90.65%, CI 0.81 to 1.0). With a probability threshold of 78.71%, 85.71% of AD patients were correctly identified (sensitivity 80.95%, specificity 92.86%).

## 3. Discussion

In the TEARAD study, AD biomarkers were tested in the tears of a cohort of 21 affected patients and 14 healthy subjects. Compared to the control group, both MCI due to AD (prodromal AD) and AD patients showed lower tear levels of Aβ1-42 and its concentration turned out to be a predictor for the disease (AUC 90.65%). In addition, affected patients showed lower CT and this turned out to be directly correlated with Aβ1-42 levels. To our knowledge, this is the first study analyzing Aβ1-42, APP-CTF, and p-tau in tear samples of a cohort of fully characterized AD patients according to the NIA-AA criteria and comparing biomarkers’ concentrations with an age- and sex-matched control group selected according to strict systemic and ocular inclusion criteria.

The idea to test AD biomarkers in tears was proposed and evaluated by a few authors without conclusive results or practical outcomes. Previous studies on AD patients’ tears are hardly comparable as they differ in patients’ baseline characteristics (i.e., criteria for AD diagnosis, systemic and ocular inclusion criteria, control group, etc.), tear collection procedure, evaluated biomarkers, as well as molecular analysis techniques [7,10]. Kalló et al. reported for the first time significant differences between total protein concentration and chemical composition in the tears of a cohort of 14 AD patients and 9 healthy controls. In particular, the combination of lipocalin-1, dermcidin, lysozyme-C, and extracellular glycoprotein lacritin levels was proposed as a potential biomarker with a specificity of 77% and a sensitivity of 81%, and lacrimal gland dysfunction in AD was speculated [11]. Kenny et al. further investigated proteins and microRNAs expression in the tears of 9 AD and 8 MCI patients compared with 15 healthy subjects [12]. They found total protein concentration in tears was similar between groups and did not show the presence of classical markers for AD in both patients and controls. However, they found a higher abundance of total microRNA in tears from AD patients with microRNA-200b-5p detected in AD samples only. Another study was conducted on 50 healthy donors and Aβ1-42 levels were found to be 10 times higher in tears than that in blood samples (approximately 10 pg/mL vs. 1 pg/mL, respectively), and inversely correlated with age [13]. The tear concentrations of Aβ1-42 in this paper are quite different from our findings in healthy subjects (10 pg/mL vs. 199.2 ± 90.3 pg/mL, respectively). Differences in tear sample collection and analysis could explain this discrepancy. Indeed, Wang and colleagues employed an electrochemical immunosensor different from the commercial antibody used in our immunoassays. Additionally, Aβ1-42 measurements from Wang and collaborators were performed requiring the addition of tear samples in a specific solution and the subsequent normalization of the data to estimate the resistance ratio. For this reason, the specific methodology as well as the adopted calculations could have impacted Aβ1-42 quantitation which cannot be compared to a quantitative method such as ELISA.

In the literature, there is only one study that investigated the presence of AD typical biomarkers in the tears of a cohort of affected patients [8]. Specifically, the study assessed AD biomarkers’ levels in tears in a cohort of 23 patients with a diagnosis of subjective cognitive decline (SCD), 22 patients with MCI, 11 patients with a diagnosis of dementia, and 9 healthy subjects. In this series, Aβ1-40 and t-tau were detectable in more than 94% of samples, while Aβ1-38, Aβ1-42, and p-tau were detectable in 18–23% of samples. In contrast to our findings, tear fluid levels of Aβ1-42 were, although not statistically significant, higher in patients compared to controls. However, results from this paper are hardly comparable with our findings for several reasons. First, in the study proposed by Gijs et al., the dementia cohort was diagnosed with the DSM-5 criteria, not including specific biomarkers for AD, and they referred to NIIA-AA criteria to characterize MCI patients with no imaging support. Second, a quantitative multi-test was used to assess AD markers levels in eluted tear samples. In addition, tear samples were collected using Schirmer strips. Conversely, we used microsponges that allowed us to directly analyze tear samples, avoiding dilution, as well as the application of normalizing formulas [14]. Third, the study population was not homogeneous with regard to age and this issue could have affected their results, as age may be correlated with biomarkers’ expression [13,15].

In our findings, significantly low tear levels of Aβ1-42 seem to be consistent with the reported pattern of biomarkers’ levels in CSF. Reduced levels of Aβ1-42 in the CSF are considered to be a result of the sequestration of peptide in the brain [16,17]. Thus, a similar mechanism involving sequestration in the lacrimal gland could occur in tears [18]. Previous studies on multiple sclerosis investigating the connections between CSF and tears reported the presence of oligoclonal bands in both samples speculating that the lacrimal gland and central nervous system’s lymphoid follicles could share similar functions [19,20]. Analogous assumptions could be formulated regarding the low Aβ1-42 levels both in CSF and tears: beta-amyloid fragments were previously identified in the lacrimal glands and, in particular, in the acinar cells [21]. We speculate that, in AD patients, beta-amyloid in tears could be low due to increased storage of the peptide fragments in the gland’s cells such as in neutrophilic granulocytes [22]. However, other cells belonging to the ocular surface could have a role in Aβ1-42 levels in tears participating in its homeostasis, therefore, future studies could bring useful data to explore this issue.

Regarding APP-CTF levels in tears, we failed to find substantial differences between patients and controls. Similarly, we did not observe any significant change in p-tau levels. Despite this, we succeeded in detecting the presence of p-tau in the tears of all included participants. This is sharply in contrast to the work from Gijs and colleagues, who did not detect p-tau in healthy controls and detected it only in 27% of demented patients. To overcome technical issues, the authors encourage the employment of other tests with higher sensitivity to detect p-tau. It is probable that the ultra-sensitive ECL reagents, as well as the different methodologies employed in our study, may have improved the ability to detect the protein in tear samples. Despite this consideration, further studies are required to confirm these findings by analyzing different techniques in parallel.

Concerning the ophthalmologic findings of our study, a reduction of CT was found in patients compared with the controls. A thinner choroid was already described in AD patients and could be associated with hypoperfusion and/or atrophic changes in the choroid, but this is the first study to explore a correlation between a biomarker in tears and the choroid [23]. In particular, as a result of the post hoc kernel regression analysis, Aβ1-42 levels in tears showed a linear correlation for CT values ranging from 170 to 260 µm. It is unlikely that Aβ1-42 levels in tears directly determine CT values or vice versa, but it is reasonable that CT reduction together with low Aβ1-42 levels in tears could both strengthen the clinical diagnosis of AD. In contrast to some previous studies, we found no substantial differences with regard to peripapillary RNFL, foveal NFL, or GPL-IPL thickness [24,25]. This could be explainable by different sample sizes and selection criteria as well as using different methods of image acquisition and measurement.

Among the strength of this study, it should be mentioned that there was a strict protocol of inclusion and exclusion criteria with well-defined criteria for MCI and AD diagnosis, age- and sex-matched population for the healthy controls, and the employment of a tear collection method as well as ELISA analysis for Aβ1-42 levels in tears that avoided excessive dilution of tear samples or normalizing formulas. Among the shortcomings, the small sample size, the non-availability of biomarkers’ analysis in the CSF, and the absence of a control group with non-AD-related dementia should be disclosed.

Overall, the results of the TEARAD study suggest that lower levels of Aβ1-42 in tears could be strongly associated with AD, pointing the way to a new promising, minimally invasive, and cost-saving method for early detection and diagnosis of AD. These findings justify future larger trials and provide useful preliminary data to plan future prospective studies to confirm the diagnostic role of Aβ1-42 in tears.

## 4. Materials and Methods

A non-pharmacological cross-sectional cohort study was conducted at the Policlinico Umberto I University Hospital of Rome from September 2020 to December 2021. The study, aiming to identify AD biomarkers in human tears (TEARAD), was performed in adherence to the tenets of the Declaration of Helsinki and was approved by the ethical board of the Sapienza University of Rome (Rif. 6090, Prot. 0027/2021). Forty right-handed patients with mild cognitive impairment (MCI) due to AD (prodromal AD) and mild-to-moderate AD attending the Department of Neurosciences and Mental Health of the Policlinico Umberto I University Hospital of Rome were assessed for eligibility to participate in the study. Parallelly, twenty age- and sex-matched volunteers with no cognitive disorders were consecutively assessed for recruitment among patients’ partners or caregivers. At the time of enrollment, all participants underwent physical and neurological examination and standard laboratory tests including serum folate, vitamin B12, and thyroid hormone assays.

General inclusion criteria for patients and controls were age between 50 and 85 years at the time of enrollment and written informed consent to the study signed by patients. To be included in this study, all the patients with prodromal AD had to meet the NIA-AA core clinical criteria for MCI due to AD, have a Clinical Dementia Rating Scale (CDR) score of 0.5 (memory box score of 0.5 or greater at screening) and a Mini-Mental State Examination (MMSE) score >24. All the participants with mild-to-moderate AD were enrolled if they met the NIA-AA core clinical criteria for probable AD, had a CDR score of 1–2 with a memory box score of at least 0.5, and a MMSE between 15 and 26 [26,27]. All patients should further have a high-resolution (3 Tesla) brain MRI with a Fazekas score of 2 or less, a Modified Hachinski Ischemic Score (MHIS) ≤ 4, an Alzheimer’s Disease Assessment Scale-Cognitive Subscale (ADAS-Cog) ≥14 and a positive florbetapir positron emission tomography (PET) showing a pathological amyloid deposition in the brain. Lastly, the key inclusion criteria for cognitive healthy controls were a CDR equal to 0, a MMSE > 26, and an ADAS-Cog < 14. Parallelly, exclusion criteria for all the participants were as follows: diabetes mellitus, uncontrolled hypertension, severe carotid artery stenosis, history of cerebrovascular disease, presence of a severe psychiatric disorder, history of repeated head trauma or protracted loss of consciousness after head trauma within the last 5 years, history of severe central nervous system (CNS) infection. Further, ophthalmological exclusion criteria comprised any ocular condition (e.g., allergic, or infectious keratoconjunctivitis) or ocular treatment potentially affecting tear and OCT analysis. In addition, patients with insufficient tear sample collection were also excluded.

### 4.1. Ophthalmological Evaluation

All participants underwent a complete ophthalmological evaluation that included medical and ophthalmic history, spherical equivalent (SEQ), lens status, IOP measured with Goldmann applanation tonometry, and anterior segment as well as fundus evaluation. According to a previously described protocol, SD-OCT (Spectralis OCT Family Acquisition Module, V 6.0.11.0 Heidelberg Engineering, Heidelberg, Germany) macular scans were automatically segmented to measure retinal thicknesses of the nerve fiber layer (NFL) and ganglion cell layer (GCL) together with inner plexiform layer (IPL) [28]. Two blinded investigators (M.G. and G.M.A.) manually corrected for any misalignment. Peripapillary Retinal Nerve Fiber Layer (pRNFL) thicknesses were acquired using a 12° circular scan centered on the optic disk with 100 frames, and 360° peripapillary RNFL, as well as superior, temporal, inferior, and nasal quadrant average thicknesses, were recorded. Choroidal thickness (CT) measurements at the central 1 mm area of the ETDRS macular grid were obtained from the horizontal, raster, 20° × 20°, 19-line enhanced depth imaging (EDI) scan protocol using a semiautomatic method [29].

### 4.2. Tear Sample Collection and Evaluation

According to a previously published protocol, tear samples were collected from affected patients and controls by microsponges applied to the lower eyelid margin of both eyes, centrifuged at 10,000 rpm per 10 min. Centrifugated tears from both eyes were pooled together and stored at −80 °C until evaluation [14]. C-terminal fragments of the amyloid precursor protein (APP-CTF), β-amyloid peptide Aβ1-42, and phosphorylated tau (p-tau) were blindly analyzed in tear samples at the Department of Biosciences and Territory of the University of Molise.

Human Aβ1-42 levels in tear samples were assessed by using Human Aβ1-42 ELISA Kit with analytical sensitivity <10 pg/mL (Thermo Fisher Scientific, Waltham, MA, USA, KHB3441), according to the manufacturer’s instructions. To assess the concentration of p-tau and APP-CTF, tear samples were prepared for Western blot analysis. To start, the method of Lowry was employed to quantify protein concentration in each sample. Subsequently, Laemmli buffer was added, and samples were denatured at 95 °C for 3 min. Protein extracts (forty micrograms of proteins) were resolved on SDS-PAGE and transferred onto nitrocellulose membrane by using a trans-blot turbo transfer system (Biorad Laboratories, Milan, Italy). Thereafter, the membrane was incubated at room temperature for 1 h with 5% no-fat dry milk in Tris-buffered saline (25 mM Tris-HCl, 138 mM NaCl, 27 mM KCl 0.05% Tween-20, pH 6.8) and probed overnight at 4 °C with the following primary antibodies: anti-tau phospho T181 (Abcam, Cambridge, UK, ab75679, dilution 1:1000); anti-β-Amyloid (Sigma-Aldrich, Milan, Italy, A8354, dilution 1:1000). Membranes were successively incubated for 1 h at room temperature with HRP-conjugated secondary anti-rabbit or anti-mouse antibodies (Biorad Laboratories, Milan, Italy). Protein-bound antibodies were visualized by Immobilon ECL Ultra Western HRP Substrate (Millipore, Burlington, NJ, USA, WBULS0100), and chemiluminescence was registered through the ChemiDoc MP system (Biorad Laboratories, Milan, Italy). Densitometric analysis derived from Western blots was then carried out by using ImageJ version 1.52t (National Institutes of Health, Bethesda, MD, USA) software for Windows. Densitometric calculations were obtained as arbitrary units (a.u.), derived from the ratio between the intensity of the protein band and the respective Ponceau S band, used as the loading control.

### 4.3. Sample Size Calculation

We based sample size calculation on the Aβ1-42 levels in tears, as it was the only biomarker that we assessed with a quantitative method (ELISA). Since no conclusive primary data regarding Aβ1-42 levels in tears was available, we assumed Aβ1-42 levels in tears to be specular to Aβ1-42 levels in CSF and 10 times less concentrated as suggested by the previous literature [13,15]. Estimating a pooled standard deviation of 50 units and an enrollment ratio of 5:3, we planned a sample size of 21 for the AD group and 13 for the control group to achieve a power of 80% and a level of significance of 5% (two-sided), to determine a true difference in means between the AD patients and the healthy subjects of 50 units. Considering 50% as average ineligibility or subsequent exclusion, we planned to assess eligibility in 40 AD patients and 24 healthy controls.

### 4.4. Statistical Analysis

Data were first analyzed by comparing the group of affected patients (MCI and AD) with that of healthy subjects to evaluate the role of the studied biomarkers to detect the affected patients. Further, to assess the ability of the potential biomarkers to discriminate the disease severity, we stratified data evaluating the differences among the three groups of enrolled patients: MCI, AD, and healthy subjects. To avoid inter-eye correlation, only one randomly selected eye of each patient was included in the statistical analysis of the ophthalmological data [30].

Statistical analysis and graph generation were performed with STATA v. 17.0 (StataCorp LLC, TX, USA). When appropriate, confidence intervals (CI 95%), interquartile range (IQR), and *p*-values were reported. The normal distribution of continuous variables was checked with the Shapiro–Wilk normality test. To test for the difference between parametric values, the unpaired t-test was employed to compare the affected patients and the control group, and the one-way ANOVA to compare three groups: patients with MCI, mild-to-moderate AD, and the control group. After ANOVA, a Tukey post hoc test was run to perform pairwise comparisons of means with equal variances between groups. To test for the differences of the non-parametric variables, the Wilcoxon rank-sum test or the Kruskal–Wallis rank test were employed accordingly. After the Kruskal–Wallis test, a Dunn’s pairwise comparison with Bonferroni adjustment was performed. Fisher’s exact test was used to compare categorical variables, and counts and percentages were reported. For parametric values, bivariate relationships were evaluated by the Pearson coefficient, while for non-parametric values, Spearman’s rank correlation was employed. To identify factors predicting AD, different from neurological scores, a logistic regression analysis was run, including sex, age, p-tau, APP-CTF, Aβ1-42, and all ophthalmological or clinical parameters that showed to be different among groups. A receiver operating characteristic (ROC) curve was generated to test the ability of the identified factors to detect AD.

## 5. Conclusions

The availability of non-invasive and inexpensive methods for an early diagnosis and/or prognosis of AD will significantly impact patients’ quality of life and health perspectives. Low levels of Aβ1-42 may represent a specific, sensitive, non-invasive, and inexpensive biomarker of AD. Given the low levels of Aβ1-42 also in the MCI, this biomarker could also represent a potential tool for early diagnosis. The direct correlation between Aβ1-42 and CT seems to strengthen the hypothesis of a direct connection between the eye and the brain. Further studies with larger sample sizes, possibly correlating tear findings to CSF and PET or MRI imaging are needed to validate our results.

## Figures and Tables

**Figure 1 ijms-24-02590-f001:**
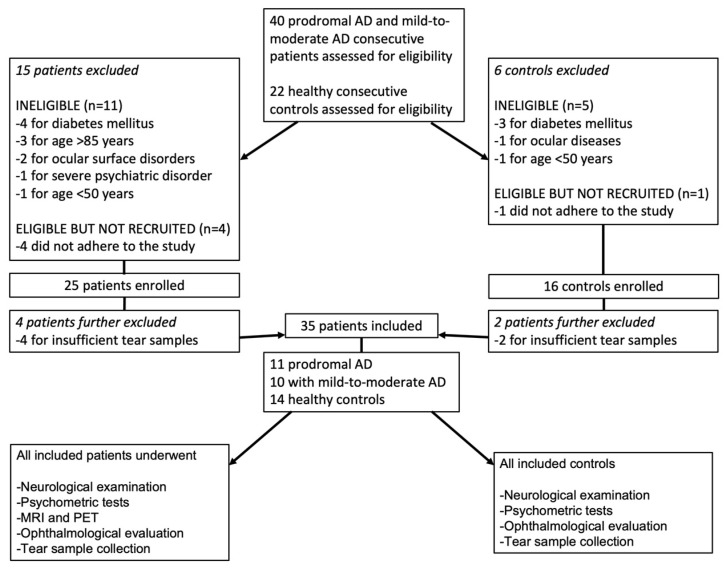
Flowchart of patients’ enrollment assessment according to the inclusion and exclusion criteria and their further exclusion as indicated in the Materials and Methods section. AD: Alzheimer’s disease, OCT: Optical coherence tomography.

**Figure 2 ijms-24-02590-f002:**
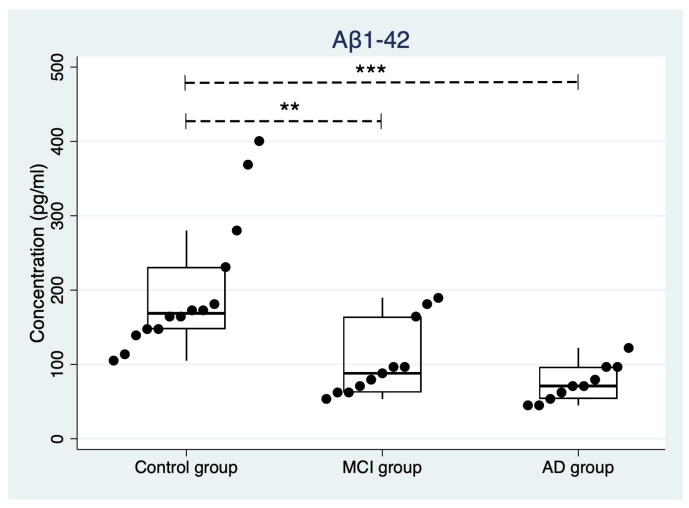
Strip plot showing the concentration of Aβ1-42 in the control group, in prodromal AD (MCI), and in mild-to-moderate AD. Boxes specify medians and quartiles, vertical spikes represent the largest or smallest value within the IQR of the upper or lower quartile. Two asterisks indicate *p* ≤ 0.01, three asterisks indicate *p* ≤ 0.001. MCI: mild cognitive impairment due to Alzheimer’s disease, AD: Alzheimer’s disease.

**Figure 3 ijms-24-02590-f003:**
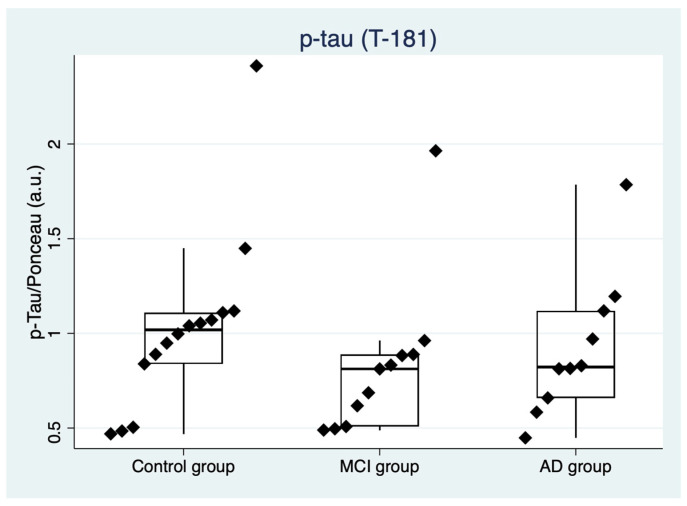
Strip plot showing the concentration of p-tau in the control group, in prodromal AD (MCI), and in mild-to-moderate AD. Boxes specify medians and quartiles, vertical spikes represent the largest or smallest value within the IQR of the upper or lower quartile. MCI: mild cognitive impairment due to Alzheimer’s disease, AD: Alzheimer’s disease.

**Figure 4 ijms-24-02590-f004:**
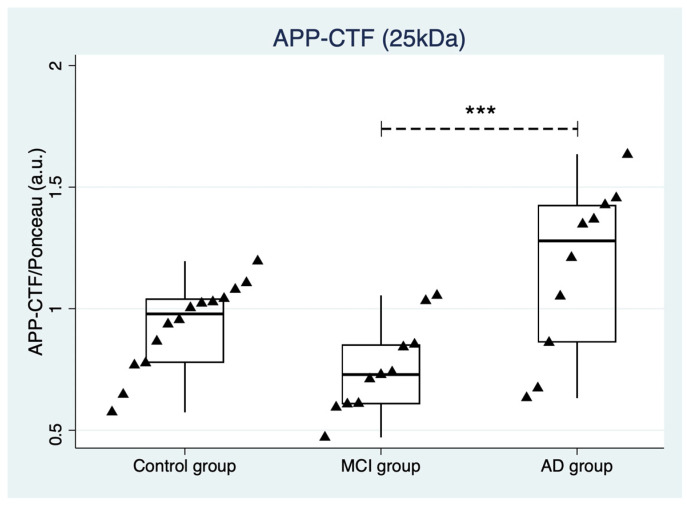
Strip plot showing the concentration of APP-CFT in the control group, in prodromal AD (MCI), and in mild-to-moderate AD. Boxes specify medians and quartiles, vertical spikes represent the largest or smallest value within the IQR of the upper or lower quartile. Three asterisks indicate *p* ≤ 0.001. MCI: mild cognitive impairment due to Alzheimer’s disease, AD: Alzheimer’s disease.

**Figure 5 ijms-24-02590-f005:**
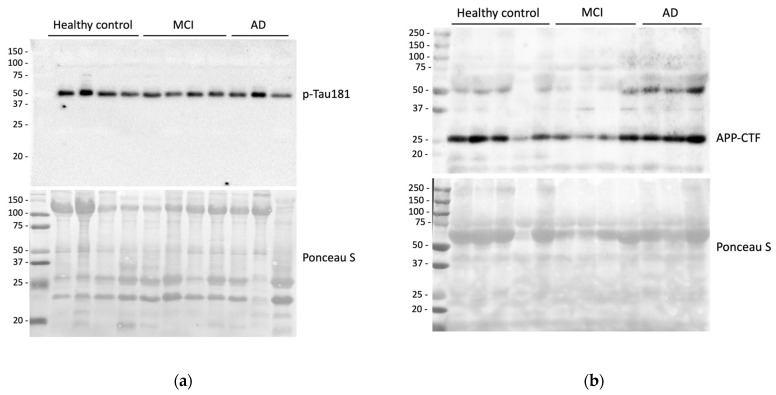
(**a**) Representative Western blot of p-tau181 in control subjects (Healthy control), mild cognitive impairment due to Alzheimer’s disease (MCI), and Alzheimer’s disease patients (AD). p-tau181 detection was carried out by probing the nitrocellulose membrane with a specific antibody, as reported in the Materials and Methods section. Ponceau S staining was employed as an internal loading control to normalize the intensity of p-tau181 bands. (**b**) Representative Western blot of APP cleavage products in control subjects (Healthy control), mild cognitive impairment due to Alzheimer’s disease (MCI), and Alzheimer’s disease patients (AD). The detection of APP cleavage products was carried out by probing the nitrocellulose membrane with a specific antibody, as reported in the Materials and Methods section. Chemiluminescence detection revealed the appearance of a prominent C-terminal fragment (APP-CTF) of approximately 25 kDa. Ponceau S staining was employed as an internal loading control to normalize the intensity of APP-CTF 25 kDa bands.

**Figure 6 ijms-24-02590-f006:**
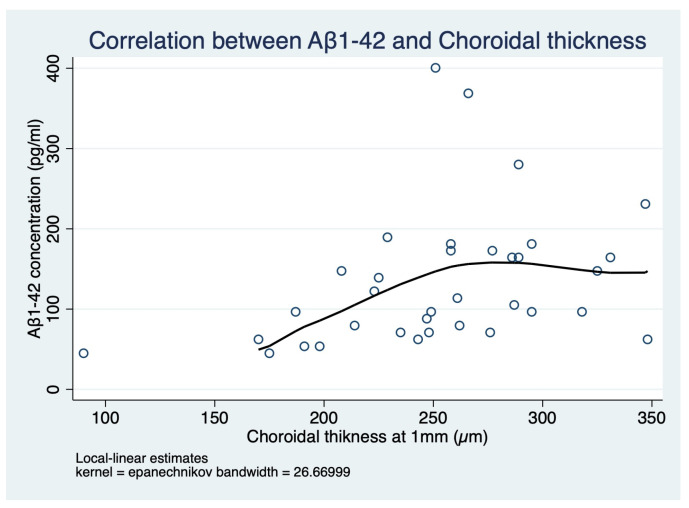
Conditional mean plot after a post hoc analysis that investigated the relationship between Aβ1-42 and choroidal thickness (CT). The graph has been generated after a non-parametric kernel regression analysis and overlays the scatterplot of the data.

**Figure 7 ijms-24-02590-f007:**
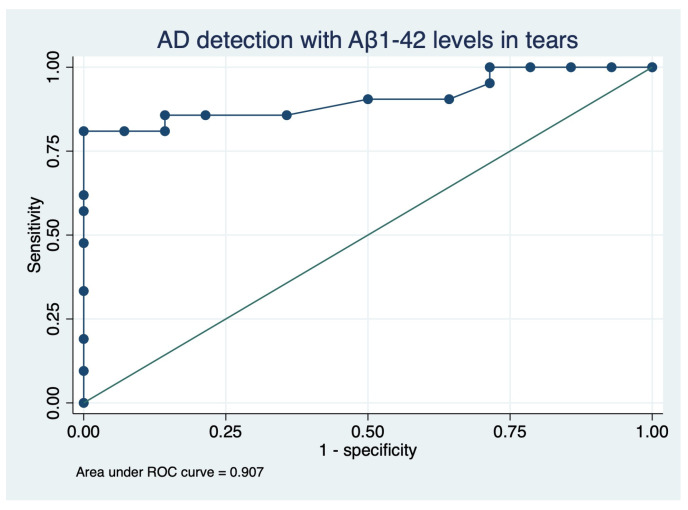
Receiver operating characteristic (ROC) curve generated after logistic regression with Aβ1-42 levels as the independent variable and the presence of AD as the dependent variable, with the area under ROC curve (AUC) of 90.65%.

**Table 1 ijms-24-02590-t001:** Demographics and clinical characteristics of enrolled patients.

Baseline Characteristics	Total	MCI and AD	CG	*p*-Value
	n 35	n 21	n 14	
Sex (female), n (%)	20 (57.1%)	11 (53.4%)	9 (64.3%)	0.728 ^‡^
Age, mean ± SD	71.7 ± 6.9	72.9 ± 7.3	69.9 ± 6.1	0.212 *
MMSE score, mean ± SD	25.9 ± 3.7	23.7 ± 3.2	29.1 ± 1.0	* **<0.001** * ^†^
ADAS-cog score, mean ± SD	18.0 ± 13.7	28.2 ± 6.4	2.6 ± 2.2	** *<0.001* ** ^†^
Systemic hypertension, n (%)	17 (48.6%)	8 (38.1%)	9 (64.3%)	0.176 ^‡^
Dyslipidemia, n (%)	15 (42.9%)	9 (42.9%)	6 (42.9%)	1.000 ^‡^
Ischemic heart disease, n (%)	3 (8.6%)	1 (4.8%)	2 (14.3%)	0.551 ^‡^
Heart arrhythmia, n (%)	1 (2.9%)	1 (4.8%)	0 (0.0%)	1.000 ^‡^
Dysthyroidism, n (%)	5 (14.3%)	1 (4.8%)	4 (28.6%)	0.134 ^‡^
COPD, n (%)	3 (8.6%)	1 (4.8%)	2 (14.3%)	0.551 ^‡^
Digestive Diseases, n (%)	4 (11.4%)	3 (14.3%)	1 (7.1%)	0.635 ^‡^

* Unpaired *t*-test, ^†^ Mann–Whitney U test, ^‡^ Fisher exact test. *p*-values < 0.05 are given in bold-italic entries. MCI: mild cognitive impairment due to Alzheimer’s disease, AD: Alzheimer’s disease. CG: control group, MMSE: Mini-Mental State Examination, ADAS-cog: Alzheimer’s Disease Assessment Scale-Cognitive Subscale, COPD: Chronic Obstructive Pulmonary Disease.

**Table 2 ijms-24-02590-t002:** Demographics and clinical characteristics stratified by mild cognitive impairment due to Alzheimer’s disease (MCI), Alzheimer’s disease (AD) patients, and control group (CG).

Baseline Characteristics	MCI	AD	MCI vs. AD	MCI vs. CG	AD vs. CG	Overall ^§^
	n 11	n 10	*p*-Value	*p*-Value	*p*-Value	*p*-Value
Sex (female), n (%)	5 (45.5%)	6 (0.6%)	0.670	0.435	1.000	0.688 ^‡^
Age, mean ± SD	75.3 ± 6.8	70.2 ± 7.1	0.202	0.122	0.991	0.108 ^†^
MMSE score, mean ± SD	25.8 ± 1.7	21.3 ± 2.7	** *0.029* **	** *<0.001* **	** *<0.001* **	***<0.001*** *
ADAS-cog score, mean ± SD	25.5 ± 6.0	31.2 ± 5.6	0.347	** *<0.001* **	** *<0.001* **	***<0.001*** *

^§^ Differences among groups (MCI, AD, and CG), * Kruskal–Wallis rank test and Dunn’s pairwise comparison, ^†^ One-way ANOVA, ^‡^ Fisher exact test. *p*-values <0.05 are given in bold-italic entries. MMSE: Mini-Mental State Examination, ADAS-cog: Alzheimer’s Disease Assessment Scale-Cognitive Subscale.

**Table 3 ijms-24-02590-t003:** Clinical and Optical coherence tomography (OCT) data of enrolled patients.

Ophthalmological Data	Total	MCI and AD	CG	*p*-Value
	n 35	n 21	n 14	
Lens status (phakic), n (%)	29 (82.9%)	18 (85.8%)	11 (78.6%)	0.664 ^‡^
SEQ, mean ± SD	−0.70 ± 1.79	−0.95 ± 1.64	−0.32 ± 2.00	0.319 *
1 mm foveal thickness, mean ± SD	272.8 ± 25.0	268.5 ± 25.4	279.1 ± 23.8	0.223 *
1 mm CT, mean ± SD	252.9 ± 54.3	237.2 ± 58.2	276.4 ± 38.9	***0.035*** *
1 mm foveal NFL, mean ± SD	12.7 ± 2.4	12.3 ± 2.3	13.4 ± 2.3	0.163 *
1 mm foveal GCL-IPL, mean ± SD	33.9 ± 8.6	32.9 ± 8.4	35.4 ± 8.8	0.405 *
Peripapillary RNFL (total), mean ± SD	92.8 ± 14.0	92.1 ± 8.8	93.9 ± 19.7	0.702 ^‡^
Peripapillary RNFL (superior), mean ± SD	115.4 ± 22.2	113.3 ± 19.6	118.4 ± 26.2	0.515 *
Peripapillary RNFL (nasal), mean ± SD	71.0 ± 13.9	71.5 ± 14.8	70.3 ± 12.8	0.800 *
Peripapillary RNFL (inferior), mean ± SD	116.1 ± 22.4	114.4 ± 17.5	118.6 ± 28.8	0.599 ^‡^
Peripapillary RNFL (temporal), mean ± SD	68.7 ± 16.0	68.9 ± 14.5	68.4 ± 18.6	0.930 *

* Unpaired *t*-test, ^‡^ Fisher exact test. *p*-values < 0.05 are given in bold-italic entries. MCI: mild cognitive impairment due to Alzheimer’s disease, AD: Alzheimer’s disease, CG: control group, SEQ: spherical equivalent, CT: choroidal thickness, NFL: nerve fiber layer, GCL-IPL: ganglion cell layer-inner plexiform layer, RNFL: retinal nerve fiber layer.

**Table 4 ijms-24-02590-t004:** Clinical and Optical coherence tomography (OCT) data stratified by mild cognitive impairment due to Alzheimer’s disease (MCI), Alzheimer’s disease (AD) patients, and control group (CG).

Ophthalmological Data	MCI	AD	MCI vs. AD	MCI vs. CG	AD vs. CG	Overall ^§^
	n 11	n 10	*p*-Value	*p*-Value	*p*-Value	*p*-Value
Lens status (phakic), n (%)	2 (18.2%)	1 (10%)	1.000 ^‡^	1.000 ^‡^	0.615 ^‡^	0.861 ^‡^
SEQ, mean ± SD	−0.67 ± 1.73	−1.25 ± 1.57	0.743	0.881	0.436	0.468 ^†^
1 mm foveal thickness, mean ± SD	278.1 ± 16.0	258.0 ± 30.2	0.146	0.993	0.096	0.082 ^†^
1 mm CT, mean ± SD	246.5 ± 39.3	227.0 ± 74.8	0.665	0.337	0.069	0.077 ^†^
1 mm foveal NFL, mean ± SD	13.0 ± 1.4	11.5 ± 2.9	0.301	0.887	0.118	0.128 ^†^
1 mm foveal GCL-IPL, mean ± SD	34.2 ± 5.3	31.4 ± 11.1	0.744	0.939	0.518	0.543 ^†^
RNFL (total), mean ± SD	90.8 ± 7.8	93.4 ± 9.9	0.716 *	0.405 *	1.000 *	0.537 *
RNFL (superior), mean ± SD	106.6 ± 12.0	120.7 ± 24.1	0.324	0.391	0.967	0.289 ^†^
RNFL (nasal), mean ± SD	71.9 ± 13.6	71.1 ± 16.8	0.991	0.957	0.990	0.961 ^†^
RNFL (inferior), mean ± SD	115.3 ± 8.3	113.5 ± 24.5	0.845 *	1.000 *	1.000 *	0.843 *
RNFL (temporal), mean ± SD	69.8 ± 17.6	67.8 ± 11.2	0.958	0.974	0.996	0.958 ^†^

^§^ Differences among groups, * Kruskal–Wallis rank test, ^†^ One-way ANOVA, ^‡^ Fisher exact test. SEQ: spherical equivalent, CT: choroidal thickness, NFL: nerve fiber layer, GCL-IPL: ganglion cell layer-inner plexiform layer, RNFL: peripapillary retinal nerve fiber layer.

**Table 5 ijms-24-02590-t005:** Alzheimer’s biomarkers’ concentration in tear samples stratified by mild cognitive impairment due to Alzheimer’s disease (MCI), Alzheimer’s disease (AD) patients, and control group (CG).

Baseline Characteristics	p-tau (a.u.)	Aβ42 (pg/mL)	APP-CTF 25 kDa (a.u.)
Total, n 35	0.94 ± 0.43	133.6 ± 84.3	0.94 ± 0.28
MCI, n 11	0.83 ± 0.41	104.1 ± 50.0	0.75 ± 0.18
AD, n 10	0.92 ± 0.38	74.3 ± 25.0	1.17 ± 0.35
CG, n 14	1.03 ± 0.49	199.2 ± 90.3	0.93 ± 0.18
MCI vs. AD, *p*-value	0.646 *	0.298 *	** *0.001* ** ^†^
MCI vs. CG, *p*-value	0.149 *	***0.008*** *	0.168 ^†^
AD vs. CG, *p*-value	0.661 *	***<0.001*** *	0.057 ^†^
MCI vs. AD vs. CG, *p*-value	0.257 *	***<0.001*** *	** *0.002* ** ^†^
MCI + AD vs. CG, *p*-value	0.148 ^§^	** *<0.001* ** ^§^	0.847 ^‡^

* Kruskal–Wallis rank test and Dunn’s pairwise comparison, ^†^ One-way ANOVA and pairwise Tukey post hoc test, ^§^ Mann–Whitney, ^‡^ Unpaired *t*-test. *p*-values < 0.05 are given in bold-italic entries. APP-CTF: C-terminal fragments of the amyloid precursor protein, p-tau: phosphorylated tau, Aβ1-42: β-amyloid peptide Aβ1-42, a.u. arbitrary unit.

## Data Availability

The data presented in this study are available on reasonable request from the corresponding author, after board approval. The data are not publicly available due to informed consent restrictions that aim to protect the privacy of research participants.

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
