# Peer review of "Beta-Amyloid Peptide in Tears: An Early Diagnostic Marker of Alzheimer’s Disease Correlated with Choroidal Thickness"

_ijms, 2023, doi:10.3390/ijms24032590_

Round 1

Reviewer 1 Report

The aim of this paper is to evaluate the diagnostic role of Alzheimer’s disease  biomarkers in tears and the possible association with alterations in retinal and choroidal microstructures. The experimental approach carried out by the authors in which they analyze the presence of biomarkers in tears is interesting from the point of view of the low invasiveness of the protocol to obtain the biological sample. My overall opinion about the data provided in this work is that they are potentially interesting. But it is necessary to review and modify the way in which they have been presented.

1) I don't understand why in some tables and in the explanation of the results, the data of the MCI and AD groups are joined and in other not. An example  are tables 3 and 4. Given the progressiveness of AD, both groups should be maintained and analyzed separately.

2) What's the informative value of the "overall" p-value shown in tables such as table 2 or table 4? Is it an average?

3) In table 3, what's the meaning of the "Total column"?

4) The results showed in table 5 and figures 2, 3, 4, and 5 need more explanations in the text.

5) The quantification of p-tau and APP-CTF is based on the intensity of a western blot for each one using the optical density of the ponceau red staining as the correction value of the protein load . The right way would be to use another protein whose levels are not affected by the experimental conditions. For example total tau and total APP.

Minor corrections

Line 25 says "affected patients" . I suppose that the authors are referring to MCI and AD patients

Line 25 "no differences were observe for APP-CTF and p-tau concentrations". Western blot don't measure concentrations

Line 43 "no definite" No definitive

In some parts of the paper, it appears "tau" in some others Tau

Author Response

We carefully read your report. Thank you for your time and for your precious and constructive comments.

Please, find below the point-to-point answers:

  1. I don't understand why in some tables and in the explanation of the results, the data of the MCI and AD groups are joined and in other not. An example  are tables 3 and 4. Given the progressiveness of AD, both groups should be maintained and analyzed separately.

Given the small sample size, we choose to analyze data by comparing affected vs healthy subjects and thereafter by evaluating MCI and AD separately. We opted for this approach to evaluate the role of the studied biomarkers to detect affected patients and within the patients’ group to discriminate disease severity. As you suggested, we clarified the rationale of our analysis in the statistical analysis section.

  1. What's the informative value of the "overall" p-value shown in tables such as table 2 or table 4? Is it an average?

In Tables 2 and 4 legends we clarify the meaning of “Overall” p-value, that stands for differences among groups.

  1. In table 3, what's the meaning of the "Total column"?

In Table 3 the “Total” column reports the results of the whole cohort.

  1. The results showed in table 5 and figures 2, 3, 4, and 5 need more explanations in the text.

In the Results’ section we report all the significant results. The main results of the study analysis were largely commented in the Discussion section. 

  1. The quantification of p-tau and APP-CTF is based on the intensity of a western blot for each one using the optical density of the ponceau red staining as the correction value of the protein load . The right way would be to use another protein whose levels are not affected by the experimental conditions. For example total tau and total APP.

Relative estimation of protein abundance by Western blot often employes an housekeeping protein, whose expression is generally not prone to modulation, which is used as an internal “control” for protein loading. For instance, GAPDH, beta-tubulin, actin, histones, and vinculin are commonly used to minimize variations related to differential protein loading across lanes. Importantly, these housekeeping proteins are nuclear, cytoplasmic or membrane proteins, thus their use is limited to studies on cell/tissue samples. Conversely, they are not appropriate for the assessment of biological fluids such as tears, blood or urine where intracellular proteins are not normally present (You et al., Exp Eye Res. 2012 Jun;99:55-62). Moreover, the abundance of protein species in biological fluids is extremely affected by the physiopathological condition, making difficult the identification of a housekeeping protein. In this context, we cannot use classic housekeeping proteins. Concerning total APP, it is an integral membrane protein, thus it is not normally found in tear fluid. Indeed, even though we used an antibody able to recognize the full-length as well as the cleavage products of APP, we only detected the 25 kDa fragment. On the other hand, total Tau could not be a good candidate as housekeeping protein, because its levels could be severely altered in dependence on the pathological condition. Coherently, it has been observed that total Tau is increased in tissues and CSF of AD patients (Sjögren et al., J Neurol Neurosurg Psychiatry. 2001 May;70(5):624-30.; Lewczuk et al., Pharmacol Rep. 2020 Jun;72(3):528-542.). In the light of the above observations, we concluded that the most accurate option to normalize the protein amount by Western blot in tears is the employment of Ponceau red, which is well-established and already used in similar experimental conditions concerning plasma and tear samples (Salvisberg et al., Proteomics Clin Appl. 2014 Apr;8(3-4):185-94; Segatto et al., Nature Communications. 2017 Nov 22;8(1):1707).

Minor corrections

Line 25 says "affected patients" . I suppose that the authors are referring to MCI and AD patients

R/ Yes we referred to MCI and AD patients.

Line 25 "no differences were observe for APP-CTF and p-tau concentrations". Western blot don't measure concentrations

R/ Amended. We changed “concentration” in “relative abundance”

Line 43 "no definite" No definitive

R/ Amended.

In some parts of the paper, it appears "tau" in some others Tau

R/ Amended.

Kind regards,

Magda Gharbiya

Reviewer 2 Report

In the current manuscript, for the first time, Magda Gharbiya and co-authors investigated the tear levels of biomarkers Aβ1-42, APP-CTF, p-Tau in healthy groups, mild cognitive impairment due to Alzheimer’s disease (MCI) and Alzheimer’s disease patients (AD) with well-defined criteria for the diagnosis. They found substantially reduced tear levels of Aβ1-42 in both MCI and AD patients, of whom choroidal thickness (CT) is also much thinner than the age and sex-matched healthy controls. Their findings indicated that the lower tear levels of Aβ1-42, with quite high specificity and sensitivity, would be a promising predictor for the early detection and diagnosis of AD as a minimally invasive and cost-saving method.

Overall, the manuscript is well-written and organized. The conclusions and figures presented in the manuscript are of excellent quality. I DO NOT HAVE ANY ISSUES OR CONCERNS.

Author Response

Thank you so much for your positive comments. Your interest encourages us to pursue this line of research and presenting a lager cohort in the future. 

In the second version of the manuscript we corrected minor typos and we extended the first paragraph of the statistical analysis as required by the other reviewer. 

Kind regards,

Magda Gharbiya

Round 2

Reviewer 1 Report

The authors have answered my questions and corrected the errors